# Disentangling the Taxonomic Status of *Caprella penantis sensu stricto* (Amphipoda: Caprellidae) Using an Integrative Approach

**DOI:** 10.3390/life12020155

**Published:** 2022-01-21

**Authors:** M. Pilar Cabezas, José M. Guerra-García, António M. Santos

**Affiliations:** 1Faculdade de Ciências da Universidade do Porto, Rua do Campo Alegre s/n, 4169-007 Porto, Portugal; amsantos@fc.up.pt; 2CIBIO-InBIO, Centro de Investigação em Biodiversidade e Recursos Genéticos, Universidade do Porto, 4485-661 Vairão, Portugal; 3Laboratorio de Biología Marina, Departamento de Zoología, Facultad de Biología, Universidad de Sevilla, 41012 Seville, Spain; jmguerra@us.es

**Keywords:** crustacea, DNA barcoding, hidden diversity, morphology, northeast Atlantic, nuclear DNA, species delimitation, Strait of Gibraltar, taxonomy

## Abstract

Despite its importance in intertidal and shallow-water marine ecosystems, *Caprella penantis* continues to be one of the most taxonomically challenging amphipods in the world. A recent molecular study focusing on *C. penantis sensu stricto* pointed out the existence of three highly divergent lineages, indicating the possible existence of a process of ongoing speciation and, thus, casting doubt on the taxonomic status of this species. In the present study, we used an integrative approach to continue to shed light on the taxonomy and distribution of this caprellid. To this end, we combined morphological and genetic data (COI and 18S) and included, for the first time, populations from its type locality. Our analyses provide strong evidence of the existence of potentially three distinct species, genetically and geographically restricted, within *C. penantis sensu stricto*, with the distribution of the true *C. penantis sensu stricto* restricted to the UK (type locality), the northern coast of the Iberian Peninsula, and the Azores. Results show the co-occurrence of two of these species in a locality of northern Portugal and indicate the existence of distinct evolutionary and diversification patterns along the eastern Atlantic region. Overall, our study highlights the use of an integrative approach to properly assess species boundaries and unravel hidden biodiversity in amphipods.

## 1. Introduction

Amphipods constitute one of the dominant taxa in shallow-water marine ecosystems worldwide. They can be easily found as epibionts of a wide variety of natural substrata, including macroalgae, hydrozoans, ascidians, sponges, corals, etc. [1,2,3,4,5], but they are also very successful colonisers of artificial substrata such as buoys, ropes, boat hulls, or pontoons, where they can reach high densities under optimal environmental conditions [6,7,8,9]. These small, benthic crustaceans form an important trophic link between primary producers and higher trophic levels [10,11,12], as they are important dietary components for many coastal marine fish species [10]. Furthermore, recent studies have shown the great usefulness of amphipods as marine bioindicators [13,14,15,16,17] and as potential resources in aquaculture [10,18,19,20,21,22].

At present, amphipods comprise more than 9000 species [23,24]. However, and despite the fact that the study of these invertebrates has increased significantly during the last few decades, amphipods remain one of the least described taxa among crustaceans [24], which means that even the most recent diversity assessments are likely underestimated [24,25,26]. In fact, a recent study indicates that about two-thirds of amphipods still remain to be discovered [24]. As with many other marine invertebrates, knowledge of amphipod biodiversity is mainly hampered by taxonomic impediment [26], but also by the realisation that cryptic species (those that are genetically distinct but lack morphological differentiation) are more common than previously anticipated [27,28,29,30]. Morphological identification of amphipod species demands the examination of numerous characters, some of which are difficult to observe since they generally display a considerable amount of intraspecific variation (e.g., ontogenetic variation and sexual dimorphism) and are not always considered in taxonomic keys (primarily based on adult males) (e.g., [28,30,31,32]). In addition, amphipods are often neglected or mistakenly identified due to their small size and the morphological uniformity among some closely related species [25,31,33,34]. Thus, an accurate morphological identification frequently requires experienced taxonomists, most of whom are nearing retirement, and who are becoming increasingly rare due to inadequate funding and the subsequent low recruitment of young scientists into this discipline [26,33,35].

Today, the use of morphological and molecular methods (e.g., DNA barcoding [36]), in the form of integrative taxonomy, has been suggested as the most effective approach for biodiversity assessments, for which molecular data provide support to accurately identify those taxa not amenable to morphological identification [37,38]. In amphipods, the use of this approach has facilitated the classification and delimitation of specimens with a more complex or hardly accessible morphology (e.g., immature or damaged specimens), allowing also the detection of putative cryptic species and species complexes within widely distributed taxa [25,29,30,39,40,41].

*Caprella penantis* Leach, 1814, is considered one of the most taxonomically challenging caprellid amphipods in the world since it has been recorded under several species or subspecies names [3,42,43,44,45]. Therefore, the need for molecular studies to determine whether it is a species complex or a single species with high intraspecific morphological variability has been highlighted since its first taxonomic studies [3,43]. In his monographs, Mayer [44,45] described 19 forms of the “*Caprella acutifrons*” group (*typica*, *minor*, *tabida*, *tibada*, *neglecta*, *gibbosa*, *andreae*, *carolinensis*, *virginia*, *lusitanica*, *natalensis*, *porcellio*, *simulatrix*, *testudo*, *angusta*, *incisa*, *verrucosa*, *borealis*, and *cristibrachium*). Several of these forms have already been given specific ranks [43,46,47,48]. For example, the last four forms were assigned to *Caprella incisa* Mayer, 1903, *C. verrucosa*, *C. borealis* Mayer, 1903, and *C. cristibrachium* Mayer, 1903, respectively [3,43,46,47]; form *andreae* was considered a distinct and valid species, *Caprella andreae* Mayer, 1890, confirmed by both morphological [3] and molecular analyses [30]. Furthermore, a recent study based on mitochondrial and nuclear markers indicated that individuals belonging to the form *gibbosa* described from Coquimbo by [44,45] could also merit specific rank [29]. Currently, only the forms *carolinensis*, *virginia*, *lusitanica*, *testudo*, and *simulatrix* are still classified as the nominal species *C. penantis sensu lato* [29,43,49], with only the last three being present in the eastern Atlantic and western Mediterranean region [29] and considered by these authors as *C. penantis sensu stricto* (hereafter *C. penantis s.s.*). Interestingly, based on molecular analyses, these authors suggest that the presence or absence of a proximal projection on the propodus of the second gnathopods, which differentiates forms *testudo* and *lusitanica* from *simulatrix*, may correspond to intraspecific variation, and thus, the form *simulatrix* should not be considered as a valid one [29]. This molecular study also revealed the existence of three highly divergent lineages in this species (*C. penantis s.s.*), suggesting the occurrence of a process of ongoing speciation and, thus, casting doubt on the taxonomic status of this marine species. Furthermore, the ultimate clarification of the taxonomical status of *C. penantis s.s.* was not possible because samples from the type locality were not available.

In this study, we used an integrative approach, combining morphological data and molecular species delimitation methods, to continue to shed light on the taxonomy and distribution of *C. penantis s.s*. To this end, we included for the first-time populations from the type locality of this species and four populations from its distributional range not previously sequenced. These data were integrated with publicly available sequences and morphological information to (1) confirm whether this caprellid is a complex of distinct species and if so, (2) examine their phylogenetic relationships; (3) determine which of them could be considered the true *C. penantis*; (4) investigate the underlying processes that lead to genetic divergence in this small marine amphipod.

## 2. Materials and Methods

### 2.1. Sample Collection and Taxonomic Identification

Between 2010 and 2018, a total of 75 specimens of the putative species *C. penantis s.s.* were by-catch in small samples of intertidal macroalgae at seven localities along the north-east Atlantic coast, including the species’ type locality (Devonshire coast) (Table 1, Figure 1). Most of the samples, except those from Viana do Castelo, were obtained as the result of a series of sampling programs conducted by different members of our research team (see Acknowledgements). At each locality, caprellids were removed by hand and immediately preserved in 96% ethanol. Sampling information, including collection localities, coordinates, and the number of individuals collected, is given in Table 1.

In the laboratory, all specimens were observed under a standard binocular microscope and identified to species level based on the morphological descriptions provided by [3,44,45,50,51]. Special attention was paid to the presence of a short and triangular rostrum on the front of the head, the shape and size of the gills, the robustness of the first antenna, the characteristics of second gnathopods in adult males, including the presence or absence of a proximal projection, and the concavity or convexity of the propodus of pereopods 5–7.

### 2.2. DNA Extraction, PCR Amplification, and Sequencing

Genomic DNA was extracted from caprellid appendages along one side of the body of each specimen, except when individuals were too small (in those cases, the whole specimen was used). We employed the Purelink Genomic DNA Mini Kit (Invitrogen, Paisley, UK) according to the manufacturer’s protocol.

A fragment (~658 bp) of the mitochondrial (mt) cytochrome c oxidase subunit I (COI) gene was amplified by polymerase chain reaction (PCR) using the primer pairs jgLCO1490/jgHCO2198 [52]. PCR amplifications consisted of 25 µL reaction volumes containing 3 µL of template DNA, 10x buffer MgCl_2_ free (Invitrogen, UK), 3 mM MgCl_2_, 0.2 mM dNTPs, 1 µM of each primer, 0.3 U Platinum Taq DNA polymerase (Invitrogen, UK) and double-distilled H_2_O to volume. PCR conditions used were as described in [53]. Furthermore, a 1051 bp fragment of the 18S ribosomal RNA (18S rRNA) nuclear gene was obtained for a subset of 24 individuals (Table 1), using the primers 18S-ai and 18S-bi [54] and following the PCR conditions specified by [29].

Positive PCR products were purified and bidirectionally sequenced by a commercial company (GENEWIZ, Leipzig, Germany).

### 2.3. Sequence Analysis

The resulting sequences were edited with Sequencher v5.4.6 (Gene Codes Corporation, Ann Arbor, MI, USA) and checked for potential contaminations using GenBank’s BLASTn search [55]. They were thereafter deposited in GenBank (Table 1).

To confirm the morphological identification of the sequenced specimens and to shed light on the taxonomy of *C. penantis s.s*., a total of 79 COI and 15 18S available sequences from this species in GenBank were included in the final data set (Appendix A, Figure 1). Furthermore, sequences of closely related species *C. dilatata* Krøyer, 1843 (24 COI; 2 18S) and *C. andreae* Mayer, 1890 (45 COI; 6 18S) were also included (Appendix A).

For the COI gene, all sequences were aligned using MUSCLE [56], as implemented in MEGA X [57]. They were also checked for the presence of pseudogenes by translating them into amino acids. On the other hand, 18S rRNA sequences were aligned using MAFFT [58], and highly variable regions and poorly aligned positions were eliminated from the analysis using Gblocks [59], with default parameters and allowing all gap positions.

### 2.4. Phylogenetic Reconstruction

Analyses were performed using data partitions by codon (1 + 2 *+* 3) for the mitochondrial COI gene, to minimise the saturation effects of codon positions on phylogenetic reconstructions [60] and to account for different rates of evolution of each one [61]. For both genes, identical sequences were collapsed into haplotypes to reduce redundancy and facilitate the computational processes, and the species *C. simia* Mayer, 1903 (accession numbers: COI—KF743437, 18S—KF743485) and *C. linearis* (Linnaeus, 1767) (accession numbers: COI—FJ581572, 18S—DQ378039) were used as outgroups.

Phylogenetic tree reconstructions were performed using two model-based methods of phylogenetic inference: maximum likelihood (ML) in Garli v2.01 [62] and Bayesian inference (BI) in MrBayes v3.2.6 [63]. The corrected Akaike Information Criterion (AICc) [64] implemented in PartitionFinder v2.1.1 [20] was used to determine the best-fit substitution model for each data set. The resulting models were GTR + I + G (1st position), F81 + I (2nd position), and GTR + G (3rd position) for the COI data set, and K80 + I for the 18S one. ML analysis was performed using 10 independent searchers and 1000 bootstrap replicates. The evaluation of log-likelihood values across searchers allowed checking the convergence between the topologies of the trees generated. The SumTrees command from the Dendropy package [65] was used to summarise non-parametric bootstrap support (BS) values for the best tree, after generating a majority-rule consensus tree. For the BI analysis, four Markov chain Monte Carlo (MCMC) chains were run twice in parallel for 2 × 10^7^ generations, sampling trees, and parameters every 1000 generations, with the heating parameter set to 0.25. Convergence of the analyses was considered to be reached when the average standard deviation of split frequencies was less than 0.01 and was validated by plotting the log-likelihood values against generation times in Tracer v1.7.1 [66]. The majority-rule consensus tree was estimated by combining results from duplicated analyses, after discarding 25% of samples as burn-in. The consensus tree inferred for each molecular data set was visualised and rooted using FigTree v1.4.4 [67] and later edited with Inkscape v1.1 software (https://www.inkscape.org, accessed on 18 December 2021). Clades with BS or BI posterior probability (BPP) greater than 70% or 0.7, respectively, were considered well supported. Furthermore, uncorrected p-distances were calculated in MEGA X [57] to estimate genetic divergences between the phylogenetic clades obtained from reconstructed trees.

Relationships between mitochondrial haplotypes were further examined by building an unrooted network in TCS v1.21 [68] following the statistical parsimony criterion [69], with a 95% connection limit. The network was plotted with tcsBU [70].

### 2.5. Molecular Species Delimitation

To evaluate the clades resulting from the phylogenetic tree reconstructions, species delimitation analyses were performed using three different approaches: two distance-based methods, the Barcode Index Number (BIN) system [71] and the Assemble Species by Automatic Partitioning (ASAP) method [72], and one tree-based method, the Bayesian Poisson Tree Process (bPTP) model [73]. All methods were applied to both COI and 18S data sets, except for the BIN system, which is based only on COI. Using BINs implemented in the Barcode of Life System (BOLD) [74], COI sequences were clustered into molecular operational taxonomic units (MOTUs) independently on their predefined taxonomic classification. Each cluster was assigned to a unique alphanumeric identifier or BIN that closely corresponds to a specific species. For the ASAP method, we applied the p-distances model implemented on the ASAP Web (https://bioinfo.mnhn.fr/abi/public/asap/asapweb.html, accessed on 19 November 2021). By building partitions from single-locus sequence alignments, it provides a score for each defined partition and sorts the sequences into putative species [72]. Finally, the bPTP model was performed on the bPTP server (https://species.h-its.org/ptp/, accessed on 19 November 2021) using the Bayesian tree as input, running 100,000 MCMC generations, and with the burn-in set to 25%. Unlike BIN and ASAP, bPTP infers putative species based on a non-ultrametric phylogenetic tree by identifying the transition points between inter- and intraspecific branching events [73].

### 2.6. Genetic Structure Analyses

To test the genetic differentiation between locations, and between regions, Fst estimations were calculated in Arlequin v3.5.1.2 [75]. For the latter, populations were grouped according to the geographical region to which they belong: continental Portugal (Alteirinhos, Viana do Castelo, Mindelo, Labruge, Castelejo, and Praia Azul) and Strait of Gibraltar (Torreguadiaro, Tarifa Island, Puerto de Ceuta, Grottes d’Hercules, Ksar-es Seghir, Estepona, El Chorrillo, Punta Carnero, Benzú, and Punta Almina) for Clade VA; UK (Sidmouth and St. Ives), northern Spain (Baleo, Cetarea, and Oyambre), the Azores (Monte da Guia), and continental Portugal (Viana do Castelo) for Clade VB. Locations with less than three individuals were excluded from the analyses. The significance of pairwise Fst values was determined by performing 10,000 permutations, under the null hypothesis of no differentiation. In addition, an analysis of molecular variance (AMOVA) [76] was performed to examine the overall genetic structure within each clade, using pairwise differences as a distance measure in Arlequin v3.5.1.2 [75].

Finally, the potential effect of isolation by distance was tested on the mitochondrial dataset applying the Mantel test to genetic and geographical distances matrices in IBD v1.52 [77] using 10,000 randomisations. Genetic distances, corresponding to pairwise Fst values, were estimated in Arlequin v3.5.1.2 [76] through 10,000 permutations, and geographic distances were assessed through the GeoDataSource platform (https://www.geodatasource.com/demo#latlong, accessed on 9 January 2022).

## 3. Results

### 3.1. Morphological Analysis

All caprellids collected were confirmed as *C. penantis s.s*. mainly based on the presence of a short and triangular rostrum on the front of the head, peduncle of antenna 1 slender, small, and elongated gills, and pereopods 5–7 propodi palm slightly concave (Figure 2) [3,50,51]. However, some small morphological differences were found among specimens from different populations (Figure 2, Appendix A). Individuals collected from the UK and the Azores were characterised by smaller and elongated gills and more elongated second gnathopods, with abundant setae and a larger proximal projection (Figure 2a,b,d), while those from continental Portugal and Morocco showed rounder and larger gills and rounder second gnathopods (Figure 2c,e). Interestingly, individuals from Viana do Castelo (Figure 2e) depicted intermediate characteristics, with more rounded second gnathopods, similar to those from Portugal and Morocco, but densely setose, similar to those from the UK and Azores populations. We were unable to include any photos of specimens from Wembury (UK) and Temara (Morocco), due to their degraded state. 

### 3.2. Phylogenetic Inference and Species Delimitation

After quality filtering, the length of the final alignment was 537 bp and 926 bp for COI and 18S, respectively. The COI data set included a total of 225 sequences, 75 from this study and 150 available from GenBank (Table 1 and Appendix A), whereas a total of 49 sequences comprised the 18S data set, 24 of them sequenced for the first time in the present study (Table 1). No stop codons, insertions, or deletions were observed.

Phylogenetic analyses of the COI data set based on BI and ML approaches, recovered trees with very similar topologies, differing only in the position of a few haplotypes within inner groups (Figure 3 and Appendix A). Both analyses highly supported the monophyly of *Caprella penantis s.s*. and the existence of three major distinct and geographically separated mtDNA clades within this species (Figure 3), henceforth named as Clade VA, VB, and VC, as this nomenclature was already established in the study of [29]. Clade VA included all haplotypes occurring in the Strait of Gibraltar and Temara (Atlantic Morocco) populations, and most haplotypes from continental Portugal (Figure 3, Table 1 and Appendix A). Clade VB grouped all haplotypes from the UK (including the species’ type locality), northern Spain, and the Azores archipelago (Figure 3, Table 1 and Appendix A). Interestingly, this clade also comprised two haplotypes (H40 and H41) observed in the Viana do Castelo population (continental Portugal). Finally, in the Clade VC, only the haplotypes present in Safi (Atlantic Morocco) were included (Figure 3, Appendix A). Genetic divergence (uncorrected p-distances) between these clades ranged from 10.2% (Clades VA and VB) to 10.6% (Clades VB and VC) (Table 2), which is similar to that found between the closely related species *C. dilatata* and *C. andreae* (11.1%). The divergence between *C. penantis* clades and the congeneric species included in the analyses were slightly higher, ranging from 13.2 to 22.6%, with the lowest value found between Clade VC and *C. dilatata*, and the highest between Clade VA and *C. linearis* (Table 2).

For the 18S gene, the ML and BI analyses rendered trees with identical topologies (Figure 4). Unlike the COI gene, only two main clades were identified within *C. penantis s.s*.—a first clade containing the haplotypes from the Safi population (Clade VC), and a second clade where the remaining haplotypes were grouped (Clade VA + VB) (Figure 4, Table 1 and Appendix A). According to these analyses, the Clade VC is sister to the species *C. andreae*, thus indicating that *C. penantis* appears to be paraphyletic, although it showed little bootstrap support (Figure 4). Within *C. penantis*, genetic divergence between the two clades was 0.4% (Table 2). The divergence between this species and the other species of the genus ranged from 0.8% (Clade VC and *C. dilatata*) to 1.6% (Clade VB and *C. simia*) (Table 2). The interspecific divergence between *Caprella* species, excluding *C. penantis*, was in the range 0.9–2.3%, with the lowest divergence found between *C. linearis* and *C. simia* and the highest between the latter and *C. andreae* (Table 2).

Finally, species delimitation analyses (BIN, ASAP, and bPTP) clustered the COI sequences of *C. penantis s.s.* in three distinct MOTUS (Figure 3), while for 18S, a total of two MOTUS were observed (Figure 4). These results agree with those from phylogenetic analyses and, therefore, strongly support the existence of at least two potentially distinct species within *C. penantis s.s*. 

### 3.3. Phylogeographic Structure

The mtDNA COI network for *C. penantis s.s*. retrieved three separate networks that could not be connected using the 95% parsimony connection limit (Figure 5). Thus, supporting the existence of three divergent groups within this species, which matched the distinct clades observed in the phylogenetic analyses (Figure 3). A total of 86 haplotypes was observed for the 154 individuals considered: 35 in Clade VA, 49 in Clade VB, and 2 in Clade VC. From these haplotypes, 79 were restricted to a single location, and only 7 were shared between 2 or more localities (up to 5) (Figure 5, Table 1 and Appendix A). The most frequent haplotypes, H23 and H59, were exclusive to Portugal and Ceuta, respectively (Figure 5, Table 1 and Appendix A). Furthermore, no central haplotype could be distinguished by higher frequencies. Within Clades VA and VB, a clear genetic and geographic structuration was observed, with many haplotypes per locality that were not shared between regions (Figure 5). Differences between haplotypes from different sampled localities were on the order of 1–19 and 1–27 mutation steps in Clade VA and Clade VB, respectively. Within Clade VA, a star-like phylogeny was recovered for Ceuta’s localities (Figure 5 and Appendix A). On the other hand, the two haplotypes conforming to the Clade VC were separated by three mutation steps (Figure 5).

Hierarchical AMOVA tests based on mitochondrial data resulted in significant genetic differences (*p* < 0.05) within populations, among populations within regions, and among geographic regions in all clades (Appendix A). In Clade VA, most of the variation (46.49%) is explained by the genetic differentiation among geographic regions: continental Portugal (Alteirinhos, Viana do Castelo, Mindelo, Labruge, Castelejo, and Praia Azul) and the Strait of Gibraltar (Torreguadiaro, Tarifa Island, Puerto de Ceuta, Grottes d’Hercules, Ksar-es Seghir, Estepona, El Chorrillo, Punta Carnero, Benzú, and Punta Almina). Observed pairwise Fst values were generally high between locations from these two regions (Fst > 0.7) but also between locations from the same region (Appendix A). In Clade VB, 22% of the variation was observed among regions (i.e., the UK, northern Spain, the Azores, and continental Portugal), but most of the variation (57.73%) was best explained by the differentiation within populations (Fst = 0.42263, *p* < 0.0001) (Appendix A). Pairwise Fst values showed that significant differentiation does not occur only among geographically distant samples (Appendix A). Overall, intermediate to large levels of divergence were found between locations from different regions and between locations within each region (Fst = 0.24696–0.70614, Appendix A).

Finally, results of the isolation-by-distance test did not show a significant correlation between genetic and geographic distances in Clade VB (r = 0.090, one-sided *p* = 0.3063). However, a significant correlation was observed in Clade VA (r = 0.5151, one-sided *p* < 0.001), thus demonstrating isolation by distance over the area studied.

## 4. Discussion

Although *Caprella* Lamarck, 1801, is by far the most species-rich genus within the family Caprellidae, with nearly 200 species distributed throughout marine ecosystems around the world [78,79], its actual diversity remains underestimated [29,30,80,81].

Our phylogenetic analyses are in agreement with those of [29], recovering three well-differentiated and supported mitochondrial lineages within *C. penantis s.s.* (Clades VA, VB, and VC) (Figure 3 and Figure 5), which were also confirmed by three approaches of species delimitation methods (Figure 3). Although the genetic divergences between these lineages were slightly lower than those observed between established caprellid species (Table 2) [29,30,81], they were within the range of COI distances reported between different peracaridean species [82,83,84,85]. Moreover, these values were 10 times greater than the mean intraspecific divergence [36] and exceeded the threshold value established for species delimitation in amphipods [83], therefore providing strong evidence of the existence of potentially three distinct species within *C. penantis s.s*. Reconstructions based on the nuclear 18S gene, however, recovered only two different lineages (Figure 4), which showed, as expected, lower values of genetic divergence (Table 2) due to the more conservative nature of this gene [86]. The slower rates of substitution observed in nuclear genes and their lower sensitivity to recent divergences than the mtDNA could explain the smaller number of lineages (and MOTUs) recovered with the 18S gene, thus indicating that Clades VA and VB are probably under an ongoing speciation process or diverged too recently to display relevant divergence in their nuclear DNA.

According to our results, it is highly likely that the true *C. penantis* corresponds to the Clade VB since it included the specimens from Devonshire coast, the type locality of this species (Figure 1, Figure 3 and Figure 5). In this clade, populations from the northern coast of the Iberian Peninsula and the Azores archipelago were also included (Figure 3 and Figure 4). Therefore, the species present in these regions correspond to the true *C. penantis s.s.*, which supports the slightly morphological affinities found between these populations (Figure 2). This species can be distinguished from the other two clades by having smaller and elongated gills and more elongated second gnathopods with abundant setae and a larger proximal projection (Figure 2, Appendix A). However, more individuals need to be morphologically analysed to determine if these characters can be used as diagnostic ones. On the other hand, Clade VA included all populations from the western and southern coast of the Iberian Peninsula, northern Morocco, and Temara (Atlantic Morocco) (Figure 1, Figure 3 and Figure 5). This clade could belong to the form *lusitanica* described from Sines (Portugal) by [44]. However, a further extensive and in-depth morphological analysis is necessary to confirm the taxonomic status of this lineage. Interestingly, our molecular and morphological results are in agreement with the morphological observations already made by Mayer [44,45], who noticed that individuals from northern Spain were morphologically more similar to those from the Azores and the UK than to those from continental Portugal. Finally, the Clade VC only included two individuals from Safi (Atlantic Morocco). Although in a previous study [29], we observed differences in the size of both individuals and gills in specimens from this population, they were not stable enough to serve as a clear diagnostic character to distinguish between lineages. Further analyses including more specimens from Safi and nearby locations are required to determine the specific status of this clade.

The disjunct distribution of these species (*C. penantis s.s.*, Clade VA, and Clade VC), indicates that allopatric speciation seems to be the most likely scenario to explain their differentiation [87,88,89]. The observed phylogeographic structure, with no shared haplotypes among populations of these species (except for Viana do Castelo) (Figure 5), was strongly reinforced by observed differences among regions and within populations in the AMOVA analysis (Appendix A). These results together with the high (for Clade VA) and moderate (for *C. penantis s.s.*) Fst values recorded (Appendix A) suggest that gene flow seems to be low among populations of these species, a feature that is not surprising for benthic, brooding organisms because of their limited capacity for autonomous dispersal [6,10]. Moreover, our results point out the existence of geographic barriers that possibly led to the genetic isolation and subsequent speciation of these species. Indeed, two main phylogeographic breaks were detected by [29] in a previous study of this species—one separating the northern portion of the Atlantic Moroccan coast from the Atlantic Iberia and the Strait of Gibraltar, and the other separating the northern Atlantic Iberia (Bay of Biscay) from the western Iberian coast (Figure 1). Although the first has also been documented in a marine isopod [90], studies focused on other peracaridean species from this region never detected any significant break [91,92]. As no other physical or ecological barriers have been reported along the northern Atlantic coast of Morocco, one hypothesis is that the differentiation of the Clade VC occurred due to vicariance events during glacial periods, as has been documented for many marine organisms [90,93], including amphipods [91,94]. Nevertheless, an extensive analysis of the genetic variation across the North African Atlantic coast is needed to shed light on this hypothesis. On the other hand, the hydrological features of the Atlantic Iberian coast, with a cooler northern region affected by upwelling and a warmer southern region with a strong Mediterranean influence [95], could explain the phylogeographic break that separates populations belonging to *C. penantis s.s.* from that belonging to the Clade VA. Populations subjected to different ecological conditions face distinct selection pressures that promote their local adaptation, preventing an effective gene flow between them which could lead to their differentiation [96]. Interestingly, these two species seem to co-occur in Viana do Castelo, as revealed by the presence of haplotypes from this location in both lineages (Figure 1, Figure 3 and Figure 5). Therefore, the possibility of sympatric speciation between *C. penantis s.s.* and Clade VA could not be ruled out. In this sense, the high morphological similarity found between these two species (Figure 2 and Appendix A), as well as their grouping in the same clade when using a nuclear gene (Figure 4), suggests that *C. penantis s.s.* and Clade VA may be the result of recent non-adaptive radiations [97] that occurred either in sympatry or in allopatry, followed by secondary contact [89,91,98]. Further morphological, molecular, ecological, and experimental studies (such as competition, predation, local adaptation, and food preferences) are needed to confirm this hypothesis and to determine if the long-term coexistence of these two species is possible. 

## Figures and Tables

**Figure 1 life-12-00155-f001:**
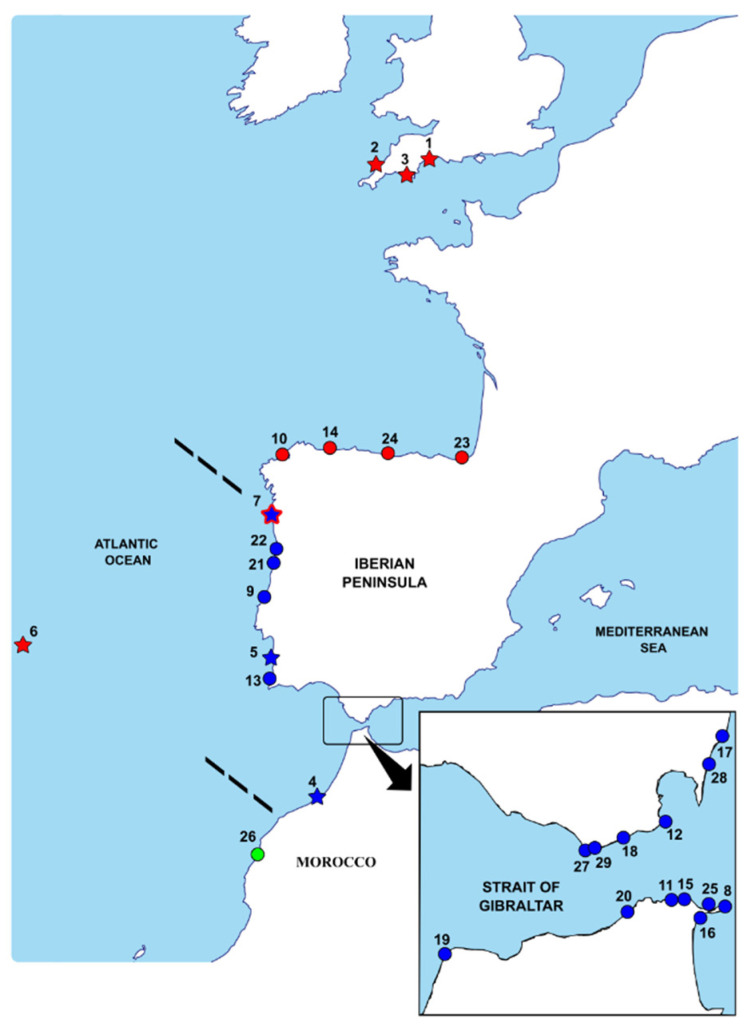
Geographic distribution of the DNA sequences included in the present study. Stars represent localities where *Caprella penantis sensu stricto* was collected and sequenced for the first time (this work), and those indicated by circles correspond to previously sequenced ones (sequences available in GenBank). See Table 1 and Appendix A for localities codes. Each locality is colour coded according to the mitochondrial clade to which they belong: blue—Clade VA, red—Clade VB, and green—Clade VC. The black dashed lines correspond to the two phylogeographic breaks detected in the present study.

**Figure 2 life-12-00155-f002:**
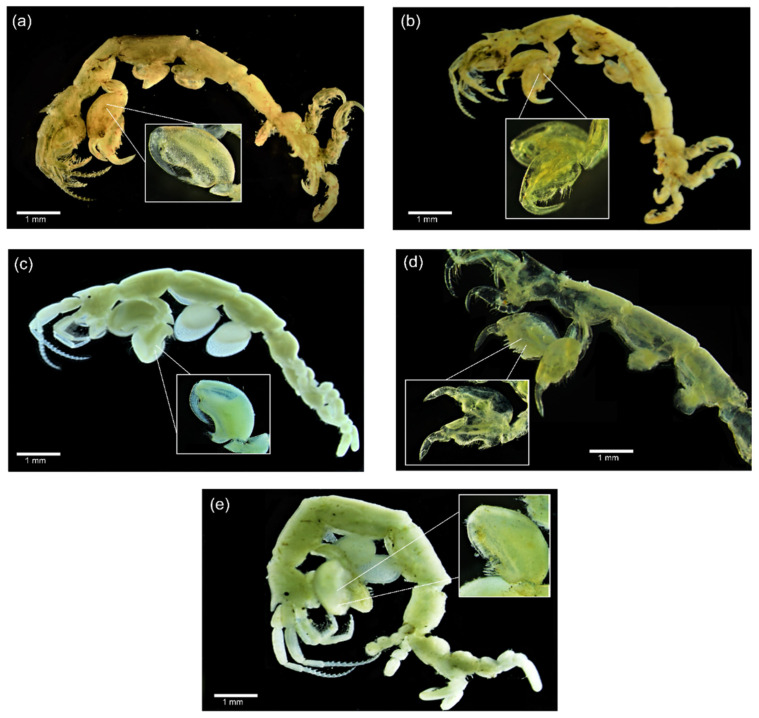
Adult male’s photographs of *Caprella penantis sensu stricto* from the localities sequenced in this study: (**a**) Sidmouth (UK); (**b**) St. Ives (UK); (**c**) Alteirinhos (continental Portugal); (**d**) Monte da Guia (Azores, Portugal); (**e**) Viana do Castelo (continental Portugal). The smaller pictures (inside a white square) represent the second gnathopods. Scale bars: 1 mm.

**Figure 3 life-12-00155-f003:**
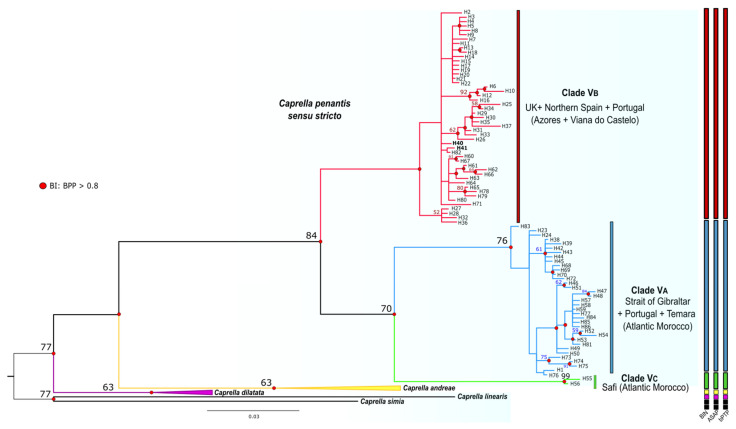
Bayesian consensus tree of *Caprella penantis sensu stricto* and related taxa based on mitochondrial COI sequences. Bayesian posterior probabilities (BPP) over 0.8 are represented by red circles at nodes and values correspond to bootstrap support (above 50) given by the maximum likelihood analyses. Clades VA–VC are identified. The tree was rooted with *C. simia* and *C. linearis* (sequences available in GenBank: KF743437 and FJ581572, respectively). The two haplotypes (H40 and H41) from Viana do Castelo grouped in Clade VB are highlighted in bold. Colour bars on the right represent results from the species delimitation analyses using Barcode Index Number (BIN), Assemble Species by Automatic Partitioning (ASAP), and Bayesian Poisson Tree Process (bPTP) models.

**Figure 4 life-12-00155-f004:**
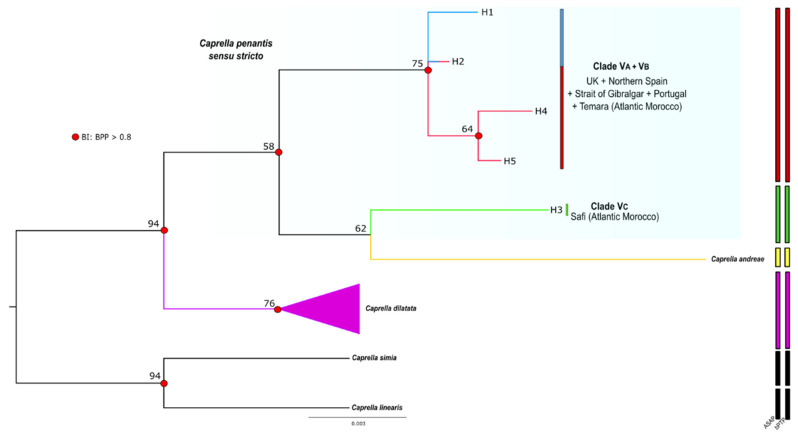
Bayesian consensus tree of *Caprella penantis sensu stricto* and related taxa based on nuclear 18S sequences. Bayesian posterior probabilities (BPP) over 0.8 are represented by red circles at nodes and values correspond to bootstrap support (above 50) given by the maximum likelihood analyses. Clades VA–VC are identified. The tree was rooted with *C. simia* and *C. linearis* (sequences available in GenBank: KF743485 and DQ378039, respectively). Colour bars on the right represent results from the species delimitation analyses: Assemble Species by Automatic Partitioning (ASAP) and Bayesian Poisson Tree Process (bPTP) models.

**Figure 5 life-12-00155-f005:**
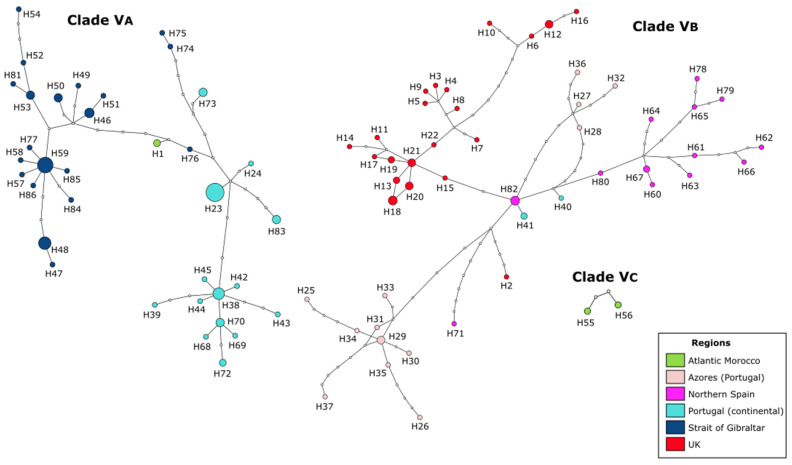
Mitochondrial COI haplotype network (95% parsimony connection limit) for *Caprella penantis sensu stricto*. Each circle represents a haplotype, and its size is proportional to the observed haplotype frequency. Colours indicate the sampling regions they belong to (see legend). Non-observed haplotypes are represented by small white circles. Every crossbeam on the connecting lines between haplotypes represents a single mutational step. Distinct clades (VA–VC) are indicated.

**Table 1 life-12-00155-t001:** List of sampling locations where *Caprella penantis sensu stricto* was collected, including codes, source countries, geographical coordinates, collection data, and number of specimens sequenced (N) for each gene (COI/18S). Genetic information regarding the number of haplotypes (H) and respective haplotype codes, the phylogenetic clade each locality belongs to, and the GenBank accession numbers for both sequenced genes are also included.

Code	Locality	Country	Coordinates	Collection Data	N	H	Haplotype Codes	Phylogenetic Clade (COI)	GenBank acc. nos. COI/18S
1	Sidmouth, Devonshire coast	UK	50°40′29.881″ N, 3°14′44.992″ W	14/08/2018	14 /3	6/1	H17-H22/H2	VB	OM057907−OM057920/OM112220−OM112222
2	St. Ives	UK	50°13′9.685″ N, 5°28′42.211″ W	13/08/2018	17/2	14/1	H3-H16/H2	VB	OM057890−OM057906/OM112218−OM112219
3	Wembury, Devonshire coast	UK	50°18′57.287″ N, 4°4′59.124″ W	20/06/2010	1	1	H2	VB	OM057889
4	Temara	Morocco	33°54′50.904″ N, 6°58′49.577″ W	31/01/2010	2/1	1/1	H1/H1	VA	OM057887−OM057888/OM112217
5	Alteirinhos	Portugal	37°31′9.89″ N, 8°47′18.949″ W	05/06/2012	12	2	H23-H24	VA	OM057921−OM057932
6	Monte da Guia, Azores	Portugal	38°31′9.102″ N, 28°37′40.778″ W	02/12/2014	15/5	13/2	H25-H37/H2	VB	OM057933−OM057947/OM112223−OM112227
7	Viana do Castelo	Portugal	41°41′59.59″ N, 8°51′24.559″ W	14/07/2014	14/13	8/1	H38-H45/H2	VA, VB	OM057948−OM057961/OM112228−OM112240

**Table 2 life-12-00155-t002:** Average *p*-distance values (%) between major mitochondrial COI (below diagonal) and nuclear 18S (above diagonal) clades identified in the phylogenetic analyses.

	Clade VA	Clade VB	Clade VC	*C. dilatata*	*C. andreae*	*C. simia*	*C. linearis*
Clade VA		--	--	--	--	--	--
Clade VB	10.2		0.4	0.9	1.1	1.6	1.3
Clade VC	10.3	10.6		0.8	1.0	1.5	1.2
*C. dilatata*	13.4	13.4	13.2		1.3	1.5	1.4
*C. andreae*	15.0	14.3	14.2	11.1		2.3	1.6
*C. simia*	19.2	19.4	19.6	17.5	17.0		0.9
*C. linearis*	22.6	21.3	21.1	19.8	20.9	19	

## Data Availability

All DNA sequences produced in the scope of this study were submitted to GenBank. The complete dataset generated during the current study is available from the corresponding author on reasonable request.

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
