# Peer review of "Disentangling the Taxonomic Status of Caprella penantis sensu stricto (Amphipoda: Caprellidae) Using an Integrative Approach"

_life, 2022, doi:10.3390/life12020155_

Round 1

Reviewer 1 Report

The work, "Unraveling the taxonomic status of Caprella penantis sensu stricto (Amphipoda:Caprellidae) using an integrative approach", it seems to me a very important contribution to the knowledge of the caprelid amphipods and in general a significant contribution to the knowledge of the peracarids; the information presented is complete and precise. The images and the tables are necessary ones and complement the information presented in the text.

I think the work is suitable for publication , I recommended a last review general.

Reviewer 2 Report

In this paper, authors rechecked taxonomic status of Caprella penenis sensu stricto using two molecular analyses (CO1 and 18s) and morphology. They concluded that there are at least 3 species based on the results. I think that methods are sound and results are clear. Therefore, I recommend to publish this paper in Life.

Minor comments

Authors suggested that allopatric speciation worked in the speciation of three species. Probably ocean currents might have driven differential locations of larvae and this would break the lineage. It would be better to see how the currents are moving and discussion on the larval dispersal in relation to the currents is involved.

Reviewer 3 Report

This manuscript tackles the issue of the species complex represented by Caprella penantis. The authors used an integrated approach coupling morphological and molecular methods to provide new insights into this taxonomic conundrum.

The manuscript is well written and scientifically sound presenting molecular data on the putative native area of the species. However, the conclusion the authors reached are almost the same as a previous paper published by some of them in the past (i.e. Hidden diversity and cryptic speciation refute cosmopolitan distribution in Caprella penantis (Crustacea: Amphipoda: Caprellidae). doi:10.1111/jzs.12010). Regardless, I believe that it deserves to be published.

I think that the morphological approach and subsequent results are lacking. The only information provided is a short description and a picture. I suggest providing (even if in the supplementary material) a table resuming the differences/similarities between clades. Also, bigger and clearer pictures are needed (the journal should not have big restrictions regarding it), if not even drawings (not mandatory if the pictures will be improved). I would expect to see the characters discerning at least C. penantis sensu stricto well presented in the main text or at least in the supplementary material.

The authors mentioned geographic barriers but did not test for any of them. Would be interesting to test for possible migration patterns inside each clade as from the haplotype networks there seems to be little on none gene flow between populations (i.e. no shared haplotype). I suggest performing at least AMOVA and isolation by distance for each clade (VA and VB). This will strengthen your results and conclusions.

MIGRATE could give also interesting results having mitochondrial and nuclear markers with patent differences, the authors could test different migration scenarios. However, this might be out of the scope of this study and it is only a suggestion that I will leave to the authors.

For further studies could be helpful to provide the final alignments in the supplementary materials as the sequences were considerably shortened before running the analyses.

Finally, if the authors have measurements, a further solution could be the morphometric approach that would also help point out the observations regarding the transition forms in Viana do Castelo.

I will leave some smaller comments for the text (mostly typos).

Line 86: The authors here mention C. penantis sensu stricto for the first time in the main text, but decided to abbreviate it only in the discussion, I would expect it here as it is mentioned several times afterwards.

Line 108: “Starts represent…” reads “Stars”.

Line 177: “Akaike Information Criterion (AICc)” should be the “corrected Akaike Information Criterion”.

Lines 178-179: “ the best-fit model of sequence evolution for each data set.” Could be better “the best-fit substitution model...”, sequence evolution sounds a bit odd.

 Lines 337-339: “The slower rates of substitution observed in nuclear genes and their less sensitivity to recent divergences than the mtDNA, could explain the lower number of lineages (and MOTUs) recovered”, “The slower rates of substitution observed in nuclear genes and their lower sensitivity to recent divergences than the mtDNA, could explain the smaller number of lineages (and MOTUs) recovered”.

Line 348: “(hereafter C. penantis s.s.)” See comment at line 86.

Line 354: “Interesting” probably “Interestingly”.

Line2 373 and 377: the citation 82 refers to isopods, not amphipods.

Round 2

Reviewer 3 Report

I appreciate the efforts f the authors and I believe the manuscript improved. I understand the issue of the morphology and I agree with the caution adopted, just be careful regarding the quality of the picture as enlarging it lost resolution (i.e. increase the resolution before enlarging it or save it already bigger).

I have only some minor comments about the AMOVA. The authors explained only the level with the highest percentage of variation explained, although all of them are significant. While in the case of Clade A this is not a problem, as there are the differences between the two regions are those with the highest divergence, in Clade B the percentage of 22 deserves to be better explored. I suggest having a look at the FST results for the pairwise comparison (and attaching them to the supplementary material) between populations/regions to have even stronger conclusions. This is because, at line 411 in the discussion, the authors wrote “These results suggest that gene flow seems to be rare or absent among populations of these species, something that is not surprising for benthic, brooding organisms because of their limited capacity for autonomous dispersal”, which is not completely true from the results as you presented. Please adjust it.

Regarding Table S2 with the morphological characters, I suggest avoiding the use of terms like “largER” or “bigER” as one would expect the second term of comparison. Please use only big, small, large, etc.

I don’t think that I will need to see again the manuscript as I am sure the authors will take care of everything. I am looking forward to seeing it published.
